# Treatments with Liquid Smoke and Certain Chemical Constituents Prevalent in Smoke Reduce Phloem Vascular Sectoriality in the Sunflower with Improvement to Growth

**DOI:** 10.3390/ijms232012468

**Published:** 2022-10-18

**Authors:** Randi Noel, Mary Benoit, Stacy L. Wilder, Spenser Waller, Michael Schueller, Richard A. Ferrieri

**Affiliations:** 1Missouri Research Reactor Center, University of Missouri, Columbia, MO 65211, USA; 2Division of Plant Sciences and Technology, University of Missouri, Columbia, MO 65211, USA; 3School of Natural Resources, University of Missouri, Columbia, MO 65211, USA; 4Chemistry Department, University of Missouri, Columbia, MO 65211, USA; 5Interdisciplinary Plant Group, University of Missouri, Columbia, MO 65211, USA

**Keywords:** sectoriality, carbon-11, liquid smoke effects, sunflower

## Abstract

Many higher plants possess a physiological organization that is based upon the carbon economy of their parts. While photosynthates are partitioned according to the relative strength of the plant’s sink tissues, in many species there is also a very close relationship between partitioning, phyllotaxy and vascular connectivity giving rise to sectorial patterns of allocation. Here, we examined the influence of smoke and certain chemical constituents prevalent in smoke including, catechol, resorcinol and hydroquinone on phloem vascular sectoriality in common sunflower (*Helianthis annuus* L.), as a model plant for sectoriality. By administering radioactive carbon-11 to a single source leaf as ^11^CO_2_, ^11^C-photosynthate allocation patterns were examined using autoradiography. A 1:200 aqueous dilution of liquid smoke treated soil caused 2.6-fold and 2.5-fold reductions in phloem sectoriality in sink leaves and roots, respectively. Treatment with catechol (1,2-d ihydroxybenzene) or resorcinol (1,3-dihydroxybenzene), polyphenolic constituents that are prevalent in smoke, caused similar reductions in phloem sectoriality in the same targeted sink tissues. However, treatment with hydroquinone (1,4-dihydroxybenzene) had no effect. Finally, the longer-term effects of smoke exposure on plant growth and performance were examined using outdoor potted plants grown over the 2022 season. Plants exposed to liquid smoke treatments of the soil on a weekly basis had larger thicker leaves possessing 35% greater lignin content than untreated control plants. They also had thicker stems although the lignin content was the same as controls. Additionally, plants exposed to treatment produced twice the number of flowers with no difference in their disk floret diameters as untreated controls. Altogether, loss of phloem sectoriality from exposure to liquid smoke in the sunflower model benefited plant performance.

## 1. Introduction

Globally, there has been increased severity in wildfires due to climate change [1]. Despite their destructive nature, often with extensive loss to personal property and land resources, fire can play an important role to restoring land resources, especially when employed in prescribed burns [2]. Most notably, it can be an important conduit for returning carbon back into the ground, in turn restoring soil health. It can also be beneficial to removing invasive plant species that might otherwise overwhelm growth of the native population [3].

Wildfires also produce an excessive amount of smoke. Studies have shown that smoke infiltration of soil can influence plant regrowth immediately following a wildfire event [4,5,6,7,8,9]. For example, seeds lying dormant for years in the soil can be suddenly stimulated to germinate upon exposure to smoke. Most notably are the so-called “fire chasers,” or ephemeral plants such as *Nicotiana attentuata* or wild tobacco that are known to spring up in great numbers throughout the Great Basin Desert regions of the U.S. after a wildfire event [5,6]. In addition to being able to break seed dormancy, certain chemical constituents within smoke have been implicated with altering root growth behavior and architecture [10,11,12,13,14,15,16,17], which in turn, can alter sink strength, or demand for photosynthates while having beneficial effects on overall plant performance. For example, increased root biomass and/or alteration of root types, have been shown to improve nutrient acquisition by better enabling plants to forage for patchy resources belowground [18].

Higher plants possess a physiological organization that is based upon the carbon economy of their parts. Here, photosynthates are partitioned across long distances according to the relative strength of the sink tissues. However, in many plant species there is also a very close relationship between partitioning, phyllotaxy and vascular connectivity giving rise to sectorial patterns of resource allocation that have evolved from selective ecological pressures over time. Such vascular restrictions in long-distance transport have been shown to exist within plants for both xylem and phloem vascular tissues [19]. Sectoriality is also not something that can be defined as being black and white, but rather as shades of gray. Many dicots differ considerably in their degree of sectoriality with natural variations occurring within both herbaceous and woody species [19,20].

One can rationalize the existence of sectoriality in nature based on its broad ecological consequences. Consider the conditions for optimal plant performance based simply on resource availability and reaction energetics. For plant performance to remain optimized, resistance to pests, diseases, or other environmental factors such as the patchy nature of available resources (e.g., light, water, and nutrients) should be variable in time and space at scales relevant to the individual plant. Take, for example, damage caused by feeding herbivores. Evidence shows that the distribution and intensity of such damage is often localized and varied in intensity across the different plant parts [21]. Mounting plant defenses often relies on the production of very specialized secondary metabolites acting as chemical defense against attack. These processes are energy demanding with repercussions to growth [22]. Logically, for a plant to mount a “full-scale” defensive response seems a waste of energy and precious resources, especially if only a portion of the plant comes under attack. Since systemic induction triggering plant defenses relies on the movement of signal molecules through the vascular system, patterns of induction will vary spatially due to vascular connectivity [23]. Hence, sectoriality from an ecological and evolutionary perspective offers a means for the plant to tailor its response to the location and severity of herbivore attack so as not to over-tax its growth resources. Even so, there are some who would argue there are benefits to a plant’s ability to mount a full-scale defensive response especially if attack is severe and long-lasting that a selective tailored response can risk catastrophic unrecoverable losses to tissues resulting in death [24]. In fact, defense induction in some sectorial plant species including tomato [25] and Arabidopsis [26] has been shown to be highly integrated across all tissues. Hence, loss of sectoriality in whole-plant defense induction might be beneficial to the plant’s fitness, especially when considering that regions populated by young developing seedlings after a fire event can be more susceptible to attack by pests.

Finally, one must also realize that the evolutionary rules of sectoriality are not cast in stone. It has been shown that manipulation of source-sink relations can break down such vascular barriers. For example, in soybean defoliation of one sector of a plant while depodding another allowed photosynthates to cross from source leaves in one sector to pods in another [27].

Here, we report on the application of using carbon-11 radiotracing coupled with autoradiography to spatially map the allocation patterns of plant carbon, in turn, examining the effects of treatments using liquid smoke on vascular sectoriality in common sunflower (*Helianthis annuus* L.) as model sectorial dicot plant [28]. Studies were expanded to include examining the individual effects of three dihydroxybenzene isomers that are prevalent in smoke including catechol, resorcinol, and hydroquinone on vascular sectoriality. We also report on the longer-term effects of chronic liquid smoke treatments on sunflower growth performance.

## 2. Results

### 2.1. Treatment with Smoke and Certain Chemical Constituents within Smoke Influence Vascular Sectoriality

Using radioactive ^11^CO_2_ (t_½_ 20.4 m) and autoradiography, we spatially mapped patterns of carbon allocation in live plants as ^11^C-photosynthates transported from a single source leaf to sink tissues (Figure 1) including young developing leaves and roots in sun flower. Growing plants in soil that was treated with liquid smoke in a laboratory setting was the closest we could come to designing a study that could mimic a post-fire event. Liquid smoke is commercially produced by condensing smoke produced from burning hardwood in cold air. Hence, the chemical constituents in smoke should be preserved in liquid smoke product. Liquid smoke treatment caused drastic reductions in phloem sectoriality. Loss of sectorialiy was observed in both sink leaves when compared to untreated control plants and (Figure 2A,B) and in roots (Figure 3A,B). Similar behavior was observed from treatments with catechol and resorcinol, but not with hydroquinone (Figure 2C–F and Figure 3C–F).

Images were quantified for the change in sectoriality and were presented in Figure 4 where a value of 1 reflected no sectoriality. Liquid smoke treatment caused 2.6-fold and 2.5-fold reductions in phloem sectoriality in sink leaves and roots, respectively (Figure 4A).

Similarly, treatment with catechol (Figure 4B) or resorcinol (Figure 4C) caused similar significant reductions in phloem sectoriality in the same targeted sink tissues. However, treatment with hydroquinone (Figure 4D) had no effect. We note that there were differences in the degree of sectoriality in control plants grown in soil versus hydrogel and attribute this slight difference in behavior to the nature of the growth media. Hence, separate controls were used in the statistical comparisons of treatment types.

### 2.2. Treatment with Liquid Smoke and Certain Chemical Constituents within Smoke Influence Root Architecture

Sunflower roots have a taproot from which grow several secondary and tertiary roots (Figure 5A,B). Liquid smoke treatment caused a significant 3.4-fold reduction in the taproot length (Figure 5C). However, this treatment had no apparent effect on total number of secondary roots, nor a significant effect on secondary root length (Figure 5D) although there was a trend of shorter roots here. Liquid smoke treatment was noted to significantly increase tertiary root length (Figure 5E) 1.6-fold relative to controls although their number was greatly diminished from an average of 57 ± 4 tertiary roots per isolated secondary root to 28 ± 5 tertiary roots per secondary root (Figure 5F).

Similar measurements were carried out on root growth as a function of treatments using catechol, resorcinol and hydroquinone (Figure 6). Treatments with catechol and resorcinol resulted in significant 7.7-fold and 3.3-fold reductions in taproot length, respectively, relative to controls while treatment with hydroquinone had no effect (Figure 6B). Catechol treatment caused a significant 1.7-fold reduction in secondary root length relative to controls, and while resorcinol treatment showed a 1.4-fold reduction in this trait, it was not significant (Figure 6C). Hydroquinone treatment had no effect on secondary root growth. Additionally, catechol and resorcinol treatments caused significant 2.8-fold and 2.2-fold increases in tertiary root length, respectively, relative to controls (Figure 6D) while hydroquinone treatment had no effect on this trait. Finally, catechol and resorcinol treatments caused significant 2.8-fold and 2.3-fold reductions in the number of tertiary roots, respectively, relative to controls (Figure 6E) while hydroquinone treatment had no effect on this trait. All together, the effects of treatments with catechol and resorcinol on root growth were very similar to those observed with liquid smoke treatments.

### 2.3. Chronic Treatment with Liquid Smoke Influences Longer-Term Plant Growth

Performance of plants grown in outdoor pots was measured weekly for 72 days (Figure 7). The number of leaves tallied each week increased exponentially (Figure 7A) with no difference observed between liquid smoke treated and untreated control plants. Leaf dimensions including leaf length (Figure 7B), leaf width (Figure 7C) and leaf thickness (Figure 7D) exhibited consistently higher values for liquid smoke treated plants than untreated control plants. Leaves of liquid smoke-treated plants also exhibited a wrinkled appearance (Appendix A) with thicker veins. Leaf length and width dimensions for both liquid smoke-treated, and untreated control plants appeared to follow a sinusoidal pattern of growth behavior with higher growth values noted during times of higher temperature. Plant height for both liquid smoke-treated and untreated control plants increased linearly over time with no difference in growth behavior between the treatment types (Figure 7E). Stem thickness increased exponentially with time where liquid smoke-treated plants consistently exhibited higher values than control plants.

The four metrics of growth showing distinct differences between liquid smoke-treated and untreated control plant data (i.e., leaf length, leaf width, leaf thickness and stem thickness) were analyzed for their relative rate of change across each sampling period (Figure 8). The relative rate of change of leaf length (Figure 8A) and leaf width (Figure 8B) showed similar trends passing through minima that coincided with lower temperature trends for the growing season. There was no difference in the growth performance of these two metrics based on their rates of change for either liquid smoke-treated or untreated control plants. The relative rate of change of leaf thickness (Figure 8C) and stem thickness (Figure 8D) also showed similar trends of exponential decrease of rates as the season progressed. There were no observed differences in these trends with the seasonal temperature change. Furthermore, there was no difference in the growth performance of these two metrics based on their rates of change for either liquid smoke-treated or untreated control plants.

At the time of the last collection of growth data, the number of flowers per plant were tallied along with measurement of their floret diameters (Figure 9). Liquid smoke treatment significantly increased flower yield 2.4-fold relative to untreated control plants, however, there was no difference in the floret diameters.

### 2.4. Treatment with Liquid Smoke Influences Lignin Content in Plant Tissue

Because liquid smoke-treated plants appeared to have “stiffer” tissues an assay for lignin content was performed on a subset of plant tissues (Figure 10). A comparison of leaves harvested from liquid smoke-treated plants had on average 35% higher lignin content relative to untreated control plants with absolute values of 50.7 ± 3.9 mg gdw^−1^ (milligram per gram of dry weight) and 72.5 ± 8.5 mg gdw^−1^ for control and liquid smoke-treated plants, respectively. Stem tissues overall had higher levels of lignin than leaves but were not different relative to treatment with absolute values of 92.3 ± 4.5 mg gdw^−1^ and 87.1 ± 2.2 mg gdw^−1^ for control and liquid smoke-treated plants, respectively.

## 3. Discussion

The present work presents clear evidence that treatment of soil with dilute liquid smoke reduces phloem vascular sectoriality in both young sink leaves and in roots of the common sunflower. It is also apparent that catechol and resorcinol chemical constituents in smoke are both bioactive causing the same effect as liquid smoke treatment, but that hydroquinone is not bioactive. Because of the structural nature of the two bioactive phenolic isomers, we suspect that they may bind to plant boron through cis-diol reactions [29] preventing this essential micronutrient from performing its important role crosslinking the pectic polysaccharide rhamnogalacturonan II in primary cell wall construction [30,31,32]. Thus, sequestration of boron by these chemical agents could weaken cell-wall integrity resulting in increased porosity [33] and perhaps “leaky’ vasculature.

Besides its role in maintaining cell-wall integrity there is mounting evidence to suggest that boron also plays a role in regulating biosynthesis and metabolism of phenolic compounds in vascular plants. Indeed, it is well known that boron deficiency causes an accumulation of phenolics through the stimulation of phenylalanine-ammonium lyase [34,35,36], as well as downstream peroxidase and oxidase enzymes [37] in the phenylpropanoid pathway producing a pool rich in polyphenolic metabolites, some of which are precursors to lignin [36,38,39]. Indeed, a key observation made in the present work was that liquid smoke treatment increased lignin content in mature sunflower leaves 35% but such treatment had no effect on the lignin content in stems.

Much of the plant’s boron, as it is taken up from the soil by the roots as boric acid, travels aboveground to the shoots through the plant’s xylem in a process mediated by the transpiration stream where boron accumulates in leaf tissues [32,40]. Once in the leaves it can be reloaded into the phloem for transport to reproductive and vegetative tissues [41,42], although this capacity varies among species [43]. Past studies in sunflower using ^10^B-labeled boric acid showed that 80% of xylem sap boron was preferentially transported to the leaves [42]. Our own past work using [^18^F]-4-fluorophenyl boronic acid as a surrogate for imaging boric acid uptake showed highest accumulation of radiotracer in the leaf edges and leaf tips of maize plants [32]. Hence, we surmise that the greatest impact bioactive chemical agents in smoke will have in creating a boron deficiency symptom is in the leaves. Of course, this is where we observed structural changes in cell-wall composition (i.e., increased lignin) as well as morphological changes in mature leaves (i.e., larger, thicker leaves with enlarged vasculature).

Loss of sectoriality can have far reaching implications to plant health. As our outdoor growth studies demonstrated, liquid smoke treated plants appeared healthier. The key variable not controlled in these studies was daily temperature. We note that trends in leaf growth, as measured by the rates of change in length and width (Figure 8), show patterns of growth that followed the weekly trends in temperature. That is, leaf growth slowed during days of lower temperature but sped up during days of higher temperature. This behavior in leaf growth has been documented across many plant species [44].

An additional metric of plant fitness can be seen from the tally of the number of flowers produced at the end of the season. Here, liquid smoke treatment caused plants to produce twice the number of flowers as untreated controls, and with no difference in flower size. Hence, we assume seed production must have been higher with liquid smoke treatment although this was not measured.

Improved fitness may be attributed, in part, to the higher lignin content in leaves providing a natural barrier to insect herbivory and/or pathogens [45]. A higher lignin content would also imply that leaves in general would have a higher phenolic content which can have certain beneficial effects in reducing herbivory by certain pests [46]. However, we must also consider that loss of sectoriality especially at an early developmental stage where integrated defense induction could be critical to a plant’s survival may be equally important. Furthermore, as plants continue to develop, their ability to uniformly allocate carbon-based resources from source leaves to all sink tissues could aid in over-all growth performance, especially if shading becomes problematic with overlapping leaves of older plants.

At this time, we cannot say whether liquid smoke treatment causes similar losses in xylem sectoriality. Evidence in the literature suggests xylem vasculature in certain higher plants can exhibit distinct connectivity giving rise to sectorial patterns in water and nutrient partitioning to aboveground tissues [47]. Future studies using radiotracers will examine this feature. We do note that observed changes in root architecture exemplified by increased tertiary root length could better enable plants to forage for sources of water and/or nutrients.

## 4. Conclusions

The present work has shown conclusively that treatment of soil with liquid smoke reduced phloem sectoriality in sunflower with improvement to plant fitness. Though speculative at this time, we suspect this may be due to *in planta* sequestration of boron by bioactive agents such as catechol and resorcinol which can impact cell-wall integrity causing a more homogeneous distribution of nutrients throughout the plant. We do not know whether changes in boron availability can alter lignin biosynthesis. As pointed out earlier, there is evidence that boron deficiency results in increased phenolics which can raise resistance to biotic stressors. More importantly, we observed increased foliar lignin content with liquid smoke treatment which also adds to improved physical barriers to pest and pathogens. However, for liquid smoke applications to become a viable farming practice, more studies are needed to determine optimal dosing and best mode of administration of the treatment. Our growth performance studies were based on repetitive dosing of pots with dilute liquid smoke. Finally, responses to treatment of the common sunflower might not be generalizable to other agriculturally relevant crops such as soybean that share similar sectorial vascular connectivity. This too requires further investigation.

## 5. Materials and Methods

### 5.1. Plant Growth for Laboratory Studies

Sunflower seeds (Mammoth variety: Ferry-Morse Home Gardening, Norton, MA, USA) were germinated and grown for 10 days in special plastic germination pouches (Phytotc, Inc., Shah Alam, Selangor, Malaysia) wetted with sterile Hoagland’s basal salt solution (PhytoTechnology Laboratories, Shawnee Mission, KS, USA). Seedlings were transferred to rhizoboxes for continued growth under 12-h photoperiods at 500 μmol m-2 s-1 light intensity, and temperatures of 25 °C/20 °C (light/dark) with relative humidity at 60% (Appendix A). Rhizoboxes were constructed using 0.635 cm thick plastic sheets (20.3 cm wide × 25.3 cm tall) sealed with high density foam gasketing (1.3 cm wide × 1.3 cm thick) that separated the front and back walls of the rhizobox (McMaster-Carr, Elmhurst, IL, USA). The entire assembly was held together with four binder clips.

For soil grown plants, soil was collected from Boone County, MO, USA. The soil belonged to the Mexico Soil Series and had a clay content of 25–35%. Soil was screened, dried and ground to a fine mesh for filling each rhizobox. To mimic the effects of smoke infiltrated soil, we used 200 mL of diluted (1:200 *v*/*v*) liquid smoke (Wright’s, Inc., Parsippany, NJ, USA) mixed with 650 cm^3^ of screened and ground soil. This level of treatment was found to stimulate seed germination in *Nicotiana attentuata* [5,6]. The contents were well-mixed and dried at 70 °C for 1-week before it was re-ground for use.

For plants grown in hydrogel we used the following procedure to generate a nutrient rich gel matrix to which 150 μM treatments of catechol, resorcinol or hydroquinone were added. Untreated hydrogels served as controls for these studies. Gels were prepared in batches starting with 3 L of deionized water to which was added 4.9 g Hoagland modified basal salt mixture (PhytoTechnology Laboratories, Shawnee Mission, KS, USA), and 1.66 g MES hydrate (Sigma-Aldrich, St. Louis, MO, USA). The pH of the solution was adjusted to 5.9 by adding 1 N sodium hydroxide solution. While stirring, 8.4 g Gelzan CM (Sigma-Aldrich, St. Louis, MO, USA) was added. The resulting solution was autoclaved (Harvey SterileMax, Thermo Fisher Scientific, Pittsburgh, PA, USA) for 15 min at 121 °C and mixed at high speed to enable aeration of the viscous solution before it set as a gel. Dihydroxybenzene treatments noted above were introduced to the solution during the aeration stage once the temperature of the mixed solution dropped below 40 °C. While still in its liquid state, contents were poured into rhizoboxes and allowed to solidify at room temperature.

### 5.2. Outdoor Plant Growth

For outdoor studies, 3 sunflower seeds were sown into a 2.7-gallon pot filled with ProMix. After germination, any excess seedlings were removed from each pot leaving a single plant. A capful of fertilizer (~1.2 g) containing nitrogen, phosphate, and potash (14-14-14, Osmocote. Smart-Release Plant Food Flower & Vegetable., The Scotts Company, Marysville, OH, USA) was added at the time of sowing. Fertilizer was reapplied to pots 30 days after germination and again at 60 days after germination. Pots were placed on elevated tables outside and were connected to drip irrigation lines which provided water twice daily in the morning and evening for a daily total of 2 L of water per pot (Appendix A). Two cohorts of plants were grown including untreated control plants (N = 23) and plants (N-22) were treated with 100 mL of diluted liquid smoke (1:200 *v*/*v* in water) weekly. Not knowing whether treatment would wash through the pots with the twice daily water sequence, we decided to expose plants to this chronic treatment schedule using diluted liquid smoke.

Plant growth performance was measured every week during the growing season. Plant height was measured with a tape measure from the soil of the pot up to the highest point of the plant in centimeters. Total leaf count was tabulated at each measurement session. Additionally, the upper most fully expanded source leaf was targeted for measurements of leaf length, width, and thickness where precision calibers (Fowler High Precision, Newton, MA, USA) were used to capture leaf thickness. Finally, stem diameters were measured using calibers 1 cm up from the top of the soil.

### 5.3. Lignin Analysis

Upper most, fully expanded source leaves were harvested from each plant at 72 days. A random subset of tissues (~100 mg) from six leaves harvested from control and smoke treated plants were freeze ground to a fine powder in 1.5 mL. Eppendorf tubes equipped with 3 mm steel balls using liquid nitrogen and a ball mill grinder (Retsch, Inc., Newtown, PA, USA). After grinding, 1.2 mL of potassium phosphate buffer (pH 7.8) mixed with 0.5% Triton was added to these tubes and they were shaken for 30 min. The tubes were then centrifuged for 10 min at 14,000× *g*. Supernatant was removed and discarded. The washing process was repeated on the pellet using the same phosphate-Triton mix, and the supernatant was again discarded after centrifugation. The pellet was then washed 3-times using 1.2 mL of methanol where the samples were vortex mixed for 3-min at each washing prior to centrifugation. The methanol was discarded after each washing cycle. The ground and washed tissue remaining was placed in a vacuum centrifuge (Labconco Centrivap Concentrator, Kansas City, MO, USA) set for 80 °C for 30-min. This process removed any remaining methanol ensuring that the samples were thoroughly dry.

Stem tissues were also harvested at the same timepoint and were removed from the lower 12 cm of the plant. Stems were dried in an oven for 1-week at 70 °C before dry grinding to a fine powder using the ball mill grinder. The same washing procedures were followed as described above.

Approximately 10 mg of the dried washed leaf or stem tissue was weighed in a 1.5 mL Eppendorf tube. A solution of acetyl bromide and glacial acetic acid (1:3 *v*/*v*) was freshly made and 250 µL of this solution was added to each weighed sample. A blank tube was also filled with this solution for later use as a reference sample on the spectrophotometer. Samples were placed in pre-heated 70 °C oven for 30 min. Upon removal, samples were immediately cooled on ice for 10-min. After cooling, 250 µL of 2 N NaOH was added to each tube. Samples were vortexed mixed and then centrifuged for 5 min at 15,000× *g*. A 250 µL aliquot of each supernatant was transferred to 5.5 mL glass vials to which 2.8 µL of hydroxylamine solution and 2.5 mL of glacial acetic acid was added and mixed well. The resulting solutions were analyzed for absorbances at 280 nm using a spectrophotometer (Thermo Scientific Evolution 201, Waltham, MA, USA). Absorbances were converted to mass units using a standard curve created from an authentic lignin standard (Sigma Aldrich, Inc., St. Louis, MO, USA).

### 5.4. Measuring Root Growth Trait

Isolated secondary roots with connected tertiary roots, harvested from rhizobox grown plants were harvested after radiotracer measurements and suspended in a tray of water. Floating the roots allowed the tertiary roots to remain isolated. Suspended roots were photographed using a digital camera. Root photographs were processed using AmScope software V4.11.18421 (AmScope, Inc., Irvine, CA, USA).

### 5.5. Production and Administration of Radioactive ^11^CO_2_

^11^CO_2_ (t_½_ 20.4 min) was produced on the GE PETtrace Cyclotron located at the Missouri Research Reactor Center using high-pressure research grade N_2_ gas target irradiated with a 16.4 MeV proton beam to generate ^11^C via the ^14^N(p,α)^11^C nuclear transformation [48,49] using our published procedures [50,51,52]. ^11^CO_2_ was administered via a leaf cell that was affixed to a targeted source leaf (Figure 1). To avoid possible effects of heterogeneous lighting on vascular sectoriality plants were uniformly illuminated with white lights at 500 μmol m^−2^ s^−1^ intensity during radiotracer administration. Once fixed, plants were incubated with radiotracer for an additional 90 min to allow transport of ^11^C-photosynthates from the targeted source leaf to sink tissues.

### 5.6. Autoradiography

After incubation with ^11^C, plants were harvested. Sink leaves were laid out and radiographic images acquired by exposing phosphor plate films for on average 10 min. For plants grown in soil, phosphor plates were laid directly onto the root-soil matrix and exposed for 40 min. For plants grown in hydrogels, the roots were gently removed and laid out on a foam imaging surface onto which the phosphor plate was placed over them. Phosphor plates were read using a Typhoon 9000 imager (Typhoon™ FLA 9000, GE Healthcare, Piscataway, NJ, USA). Images were quantified using ImageQuant™ TL 7.0 software (Cytiva Life Sciences, Inc., Marlborough, MA, USA) to determine the extent of sectoriality both within sink leaves and in roots. For leaf tissue, levels of radioactivity were determined for the left-side and right-side of each sink leaf using the midrib as the dividing line between halves. Similarly, for quantifying root images, the tap root was used as the dividing line distinguishing secondary and tertiary roots on the left-side and right-side of the tap root.

### 5.7. Statistical Analysis

Data was analyzed using the student’s *t*-test for pair-wise comparisons made between non-treated growth media (either soil or hydrogel) representing control plants and growth media that was treated with either liquid smoke, or with certain chemicals prevalent in smoke including catechol, resorcinol, and hydroquinone. Statistical significance was set at *p* < 0.05.

## Figures and Tables

**Figure 1 ijms-23-12468-f001:**
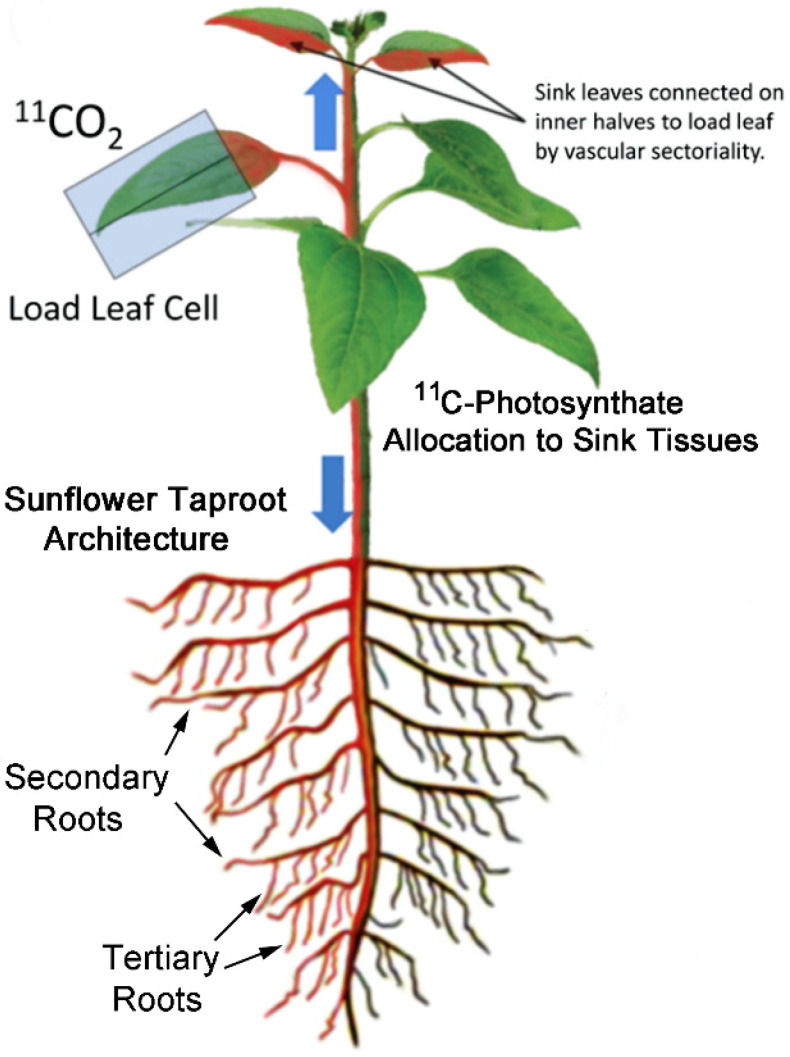
Experimental setup showing patterns of ^11^C-allocation. ^11^CO_2_ was administered to a single source leaf encased in a gas tight cell. Export ^11^C-photosynthates to sink tissues (i.e., young leaves and roots) is shown in red with direct connections to only half of two younger sink leaves attached immediately above that source leaf and half the root system.

**Figure 2 ijms-23-12468-f002:**
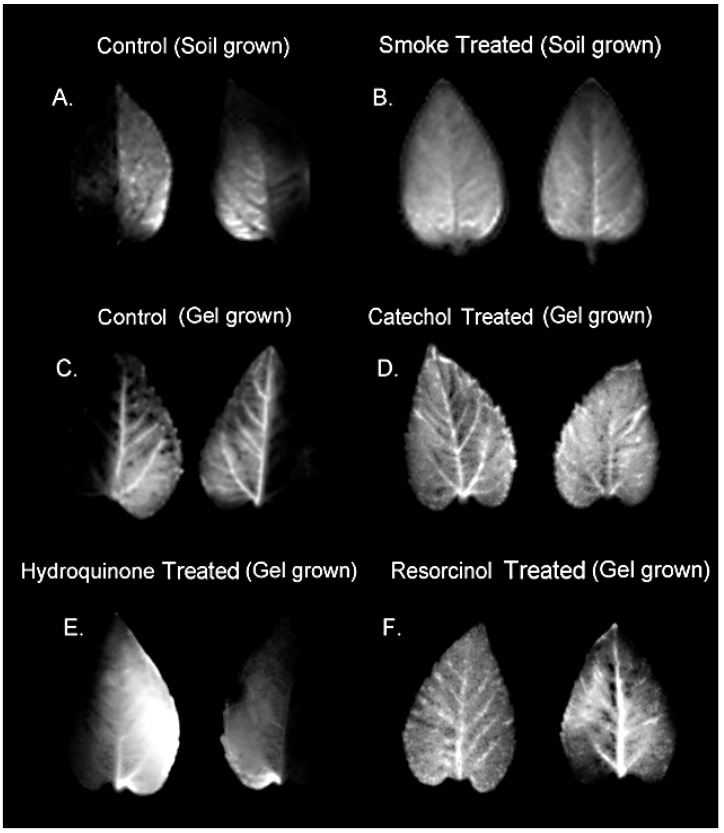
Radiographic images showing patterns of ^11^C-photosynthate allocation to sink leaves. Panel (**A**): control sink leaves of plants grown in soil show the characteristic pattern of sectorial vascular connectivity. Panel (**B**): sink leaves from plants grown in liquid smoke treated soil show loss of sectorial connectivity. Panel (**C**): control sink leaves of plants grown in a hydrogel matrix show similar sectorial behavior as soil grown plants. Panel (**D**): sink leaves of plants grown in a hydrogel matrix treated with 150 μM catechol show loss of sectoriality. Panel (**E**): sink leaves of plants grown in a hydrogel matrix treated with 150 μM hydroquinone show similar sectorial patterns as control plants. Panel (**F**): sink leaves of plants grown in a hydrogel matrix treated with 150 μM resorcinol show loss of sectoriality.

**Figure 3 ijms-23-12468-f003:**
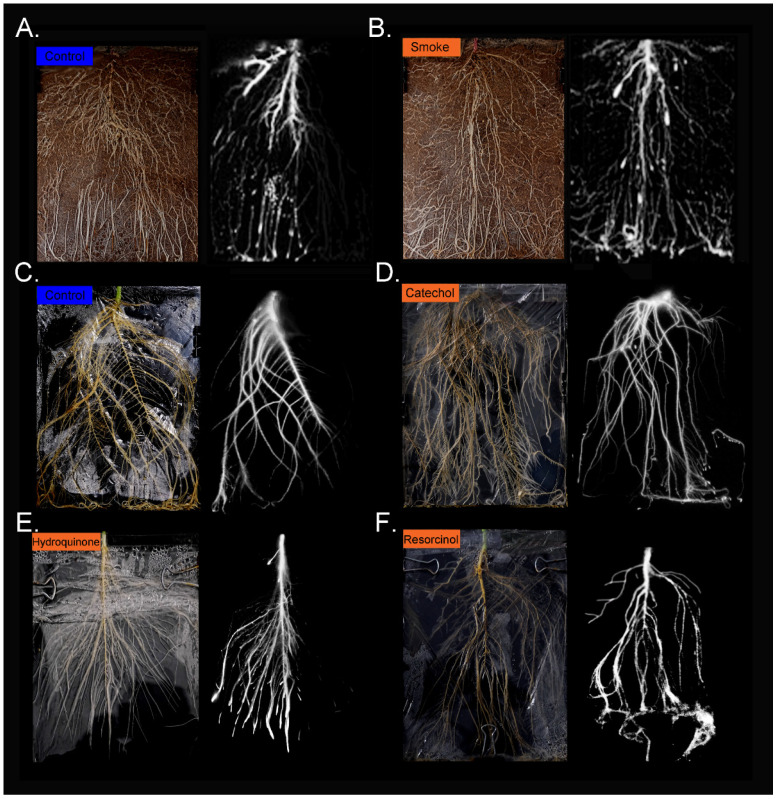
Root photographs with matching radiographic images showing patterns of ^11^C-photosynthate allocation to sink roots. Panel (**A**): control sink roots of plants grown in soil show the characteristic pattern of sectorial vascular connectivity. Panel (**B**): sink roots from plants grown in liquid smoke treated soil show loss of sectorial connectivity. (Panel (**C**)): control sink roots of plants grown in a hydrogel matrix show similar sectorial behavior as soil grown plants. Panel (**D**): sink roots of plants grown in a hydrogel matrix treated with 150 μM catechol show loss of sectoriality. (Panel (**E**)): sink roots of plants grown in a hydrogel matrix treated with 150 μM hydroquinone show similar sectorial patterns as control plants. Panel (**F**): sink roots of plants grown a hydrogel matrix treated with 150 μM resorcinol show loss of sectoriality.

**Figure 4 ijms-23-12468-f004:**
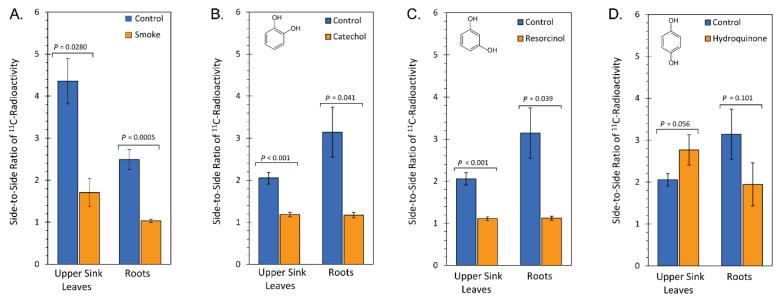
Bar graphs showing changes in vascular sectoriality as a function of treatment. The degree of sectoriality is depicted as the side-to-side ratio of ^11^C-radioactivity measured from radiographic images of leaves and roots, (**A**) liquid smoke treatment, (**B**) treatment with catechol, (**C**) treatment with resorcinol, (**D**) treatment with hydroquinone. The midrib of the leaf and taproot were used as the dividing lines for activity measurements using ImageQuant™ TL 7.0 software (Cytiva Life Sciences, Inc., Marlborough, MA, USA) applied to ^11^C-radiographs. A value of 1 in the graph reflects no sectoriality. Data represents means ± SE for N = 4 replicates where *p* < 0.05 were statistically significant.

**Figure 5 ijms-23-12468-f005:**
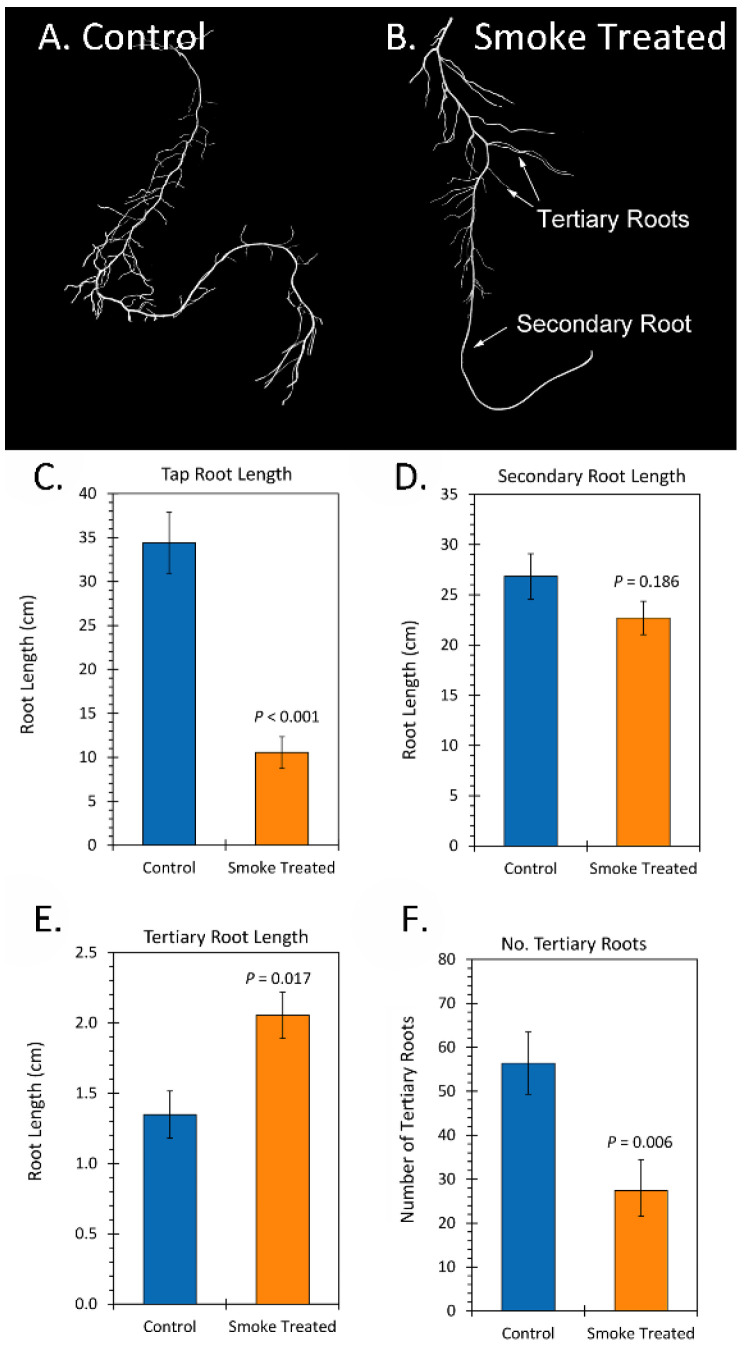
Effect of liquid smoke treatment on sunflower root architecture. Panel (**A**,**B**): Digital photographs of floated isolated roots from rhizobox studies. Panel (**C**): taproot length in centimeters measured from full rhizobox digital images taken during studies (see Figure 3A,B). Panel (**D**): secondary root length in centimeters measured digitally from floated root images. Panel (**E**): tertiary root length in centimeters measured digitally from floated root images. Panel (**F**): number of tertiary roots tallied per isolated secondary root. All graphical data bars reflect average values ± SE (N = 4). *p*-values < 0.05 were considered statistically significant.

**Figure 6 ijms-23-12468-f006:**
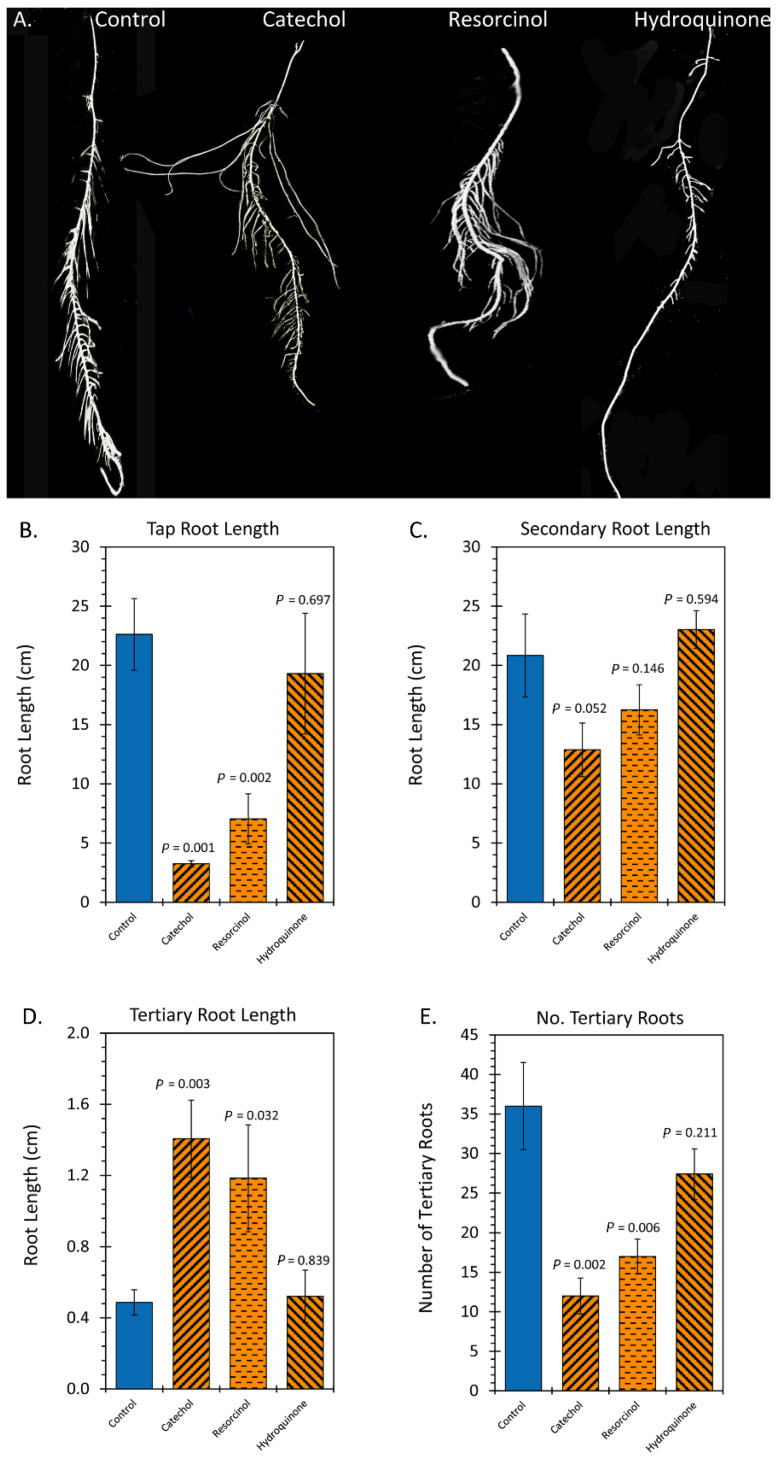
Effect of chemical treatments on sunflower root architecture. Panel (**A**): Digital photographs of floated isolated roots from rhizobox studies from control, catechol, resorcinol, and hydroquinone treated plants. Panel (**B**): taproot length in centimeters measured from full rhizobox digital images taken during studies (see Figure 3C–F). Panel (**C**): secondary root length in centimeters measured digitally from floated root images. Panel (**D**): tertiary root length in centimeters measured digitally from floated root images. Panel (**E**): number of tertiary roots tallied per isolated secondary root. All graphical data bars reflect average values ± SE (N = 4). *p*-values < 0.05 were considered statistically significant.

**Figure 7 ijms-23-12468-f007:**
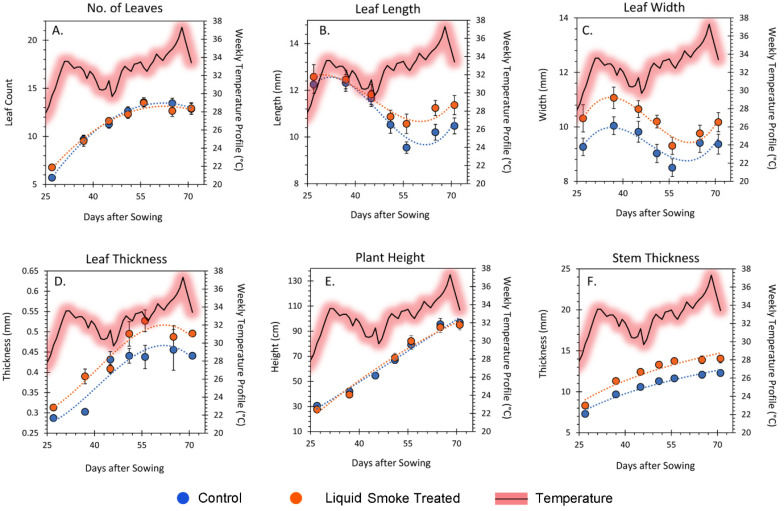
Plant growth performance for outdoor potted plants. Panel (**A**): total number of plant leaves were tallied weekly for liquid smoke treated and untreated control plants. Panel (**B**–**D**): upper most fully expanded source leaf dimensions were measured in millimeters (mm) which included leaf length, leaf width and leaf thickness. Panel (**E**): plant height was measured in centimeters (cm) and monitored weekly. Panel (**F**): stem thickness was measured in millimeters (mm). Each data point represents an average value ± SE for control, N = 22 plants and liquid smoke-treated, N = 23 plants. Data were fit to best trends shown in each panel as the dashed lines. Furthermore, each panel shows the temperature trend that was recorded over the growing season for the 5-day average of temperature highs.

**Figure 8 ijms-23-12468-f008:**
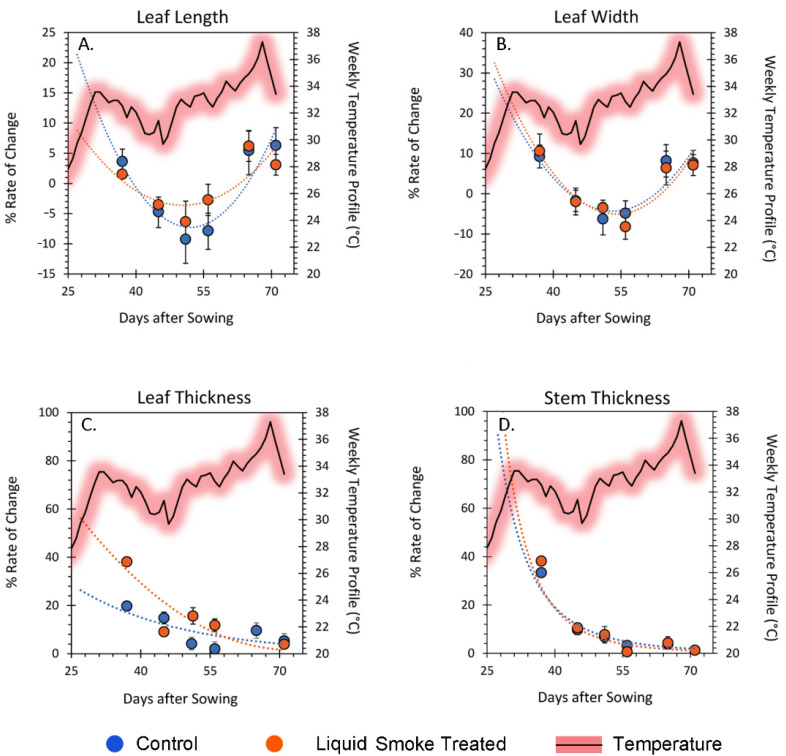
Relative change in plant growth for outdoor potted plants. Panel (**A**): relative change in leaf length growth. Panel (**B**): relative change in leaf width. Panel (**C**): relative change in leaf thickness. Panel (**D**): relative change in stem thickness. Each data point represents average values ± PE (propagated error) for control, N = 22 plants, and liquid smoke-treated, N = 23 plants. Data were fit to best trends shown in each panel as the dashed lines. Each panel also shows the temperature trend over the growing season for the 5-day average of temperature highs.

**Figure 9 ijms-23-12468-f009:**
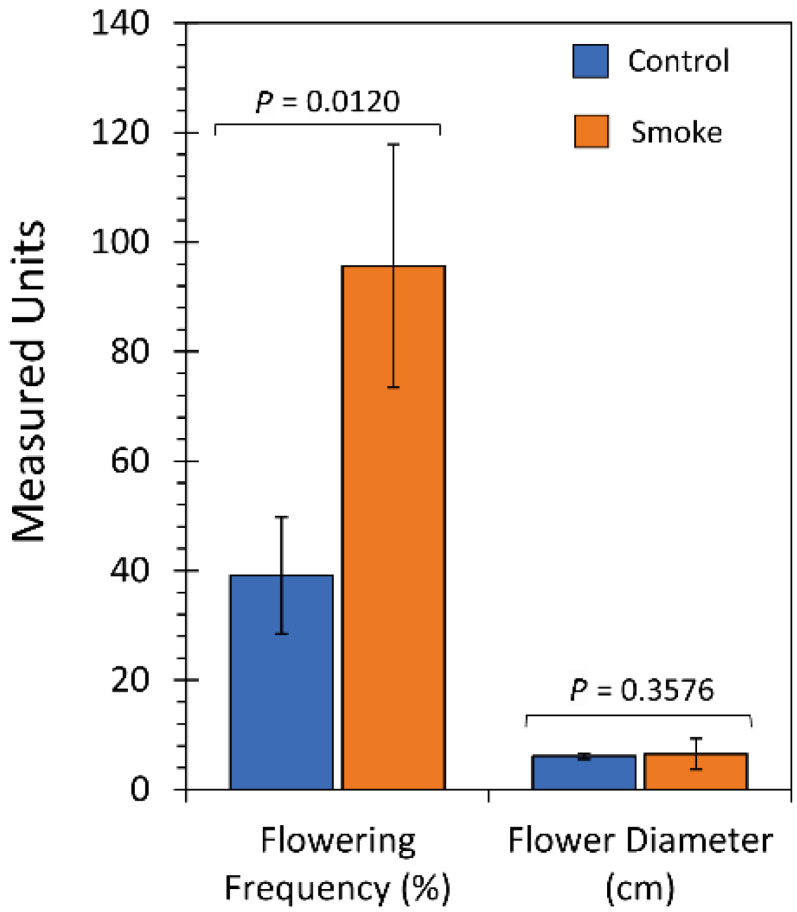
Flowering metrics. The relative flowering frequency was measured at 72 days after sowing. This metric accounted for multiple flowers on individual plants. Flower diameter, measured in centimeters (cm), represented the diameter of the disk floret. Data bars represent average values ± SE for control, N = 22 plants and liquid smoke-treated, N = 23 plants. *p*-values < 0.05 were considered statistically significant.

**Figure 10 ijms-23-12468-f010:**
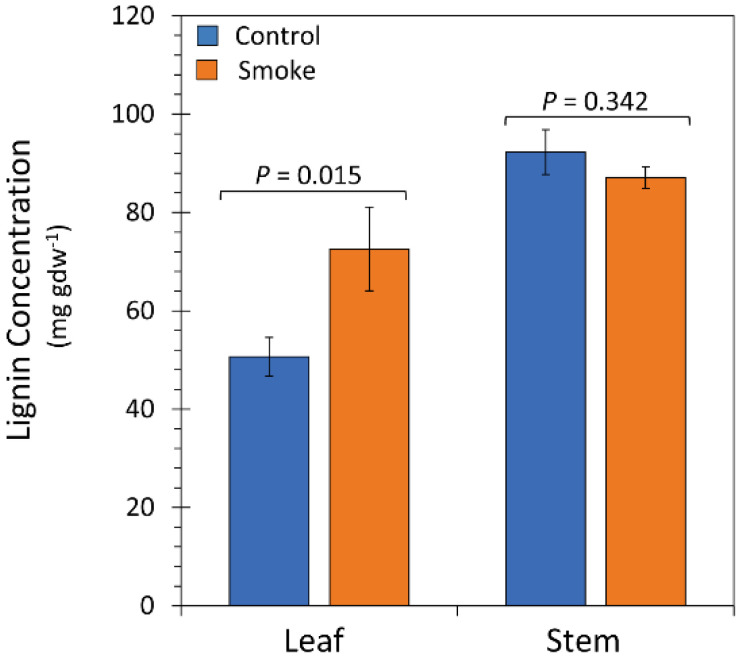
Lignin content. Leaf and stem lignin concentrations are presented in milligrams lignin per gram dry weight of tissue (mg gdw^−1^). Data bars represent average values ± SE, N = 6 leaves for control and smoke treated, N = 5 stems for control and N = 6 stems for liquid smoke-treated. *p*-values < 0.05 were considered statistically significant.

## Data Availability

Data will be made available upon request to corresponding author.

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
