# Peer review of "Treatments with Liquid Smoke and Certain Chemical Constituents Prevalent in Smoke Reduce Phloem Vascular Sectoriality in the Sunflower with Improvement to Growth"

_ijms, 2022, doi:10.3390/ijms232012468_

Round 1

Reviewer 1 Report

I found the manuscript to be well written, clear in its statements, and conclusive in its discussion. It is well organized and the data supports the conclusion drawn by the authors. I only have a few minor points to criticize, one of which is that the title and the text talk about the effects of smoke on on plants. However, no actual smoke was used; rather liquid smoke was used to treat plants. This further raises the question why real smoke was not used. I am aware that this might be impossible, but for the reader the manuscript should clearly stated that a.) no real smoke was used, and b.) why is the use of liquid smoke appropriate in this context. Also, the authors say that they used smoke-infiltrated soil, when in fact soil treated with liquid smoke was used. Also in this context, why was the soil repeatedly treated with liquid smoke? If a fire goes through, how long does smoke remain in the soil? And is this why the soil was treated for several days with the liquid smoke? This needs some clarification with particular emphasis on the relevance of the treatments. 

Author Response

For clarity I provided the reviewer's text followed my my responses in boldfaced italics.

I found the manuscript to be well written, clear in its statements, and conclusive in its discussion. It is well organized, and the data supports the conclusion drawn by the authors. I only have a few minor points to criticize, one of which is that the title and the text talk about the effects of smoke on plants. However, no actual smoke was used; rather liquid smoke was used to treat plants. Please note that the title has been changed to reflect the fact that we were using liquid smoke in our treatments.  Further, we edited any text within the manuscript that stated, “smoke infiltrated soil” replacing it with “liquid smoke-treated.”

This further raises the question why real smoke was not used. I am aware that this might be impossible, but for the reader the manuscript should clearly state that a.) no real smoke was used, and b.) why is the use of liquid smoke appropriate in this context. Liquid smoke is a commercially produced product made by burning hardwood and condensing the smoke in cold air. Many researchers use this product to study seed germination. Note that in lines 111-115 in the edited manuscript we describe what liquid smoke is and how it is produced.  This should provide clarity now for readers.

Also, the authors say that they used smoke-infiltrated soil, when in fact soil treated with liquid smoke was used. Please see our response above indicating that all text in the manuscript was edited to reflect the fact that we used liquid smoke treatments.

Also in this context, why was the soil repeatedly treated with liquid smoke? If a fire goes through, how long does smoke remain in the soil? And is this why the soil was treated for several days with the liquid smoke? This needs some clarification with particular emphasis on the relevance of the treatments. There is not much information regarding how deep smoke will penetrate the soil column nor about how long chemicals from smoke remain in the soil.  We were worried that in our outdoor potted plant studies that the twice daily irrigation of pots would wash away the chemical constituents in liquid smoke treatment.  Hence, we decided to examine plant growth performance using chronic diluted liquid smoke treatments administered weekly throughout the growing season. Please see lines 382-385 in the Methods section.

Reviewer 2 Report

Review ID ijms-1943563

 Rising up from the Ashes: Smoke and Certain Chemical Con stituents Prevalent in Smoke Reduce Vascular Sectoriality in the Sunflower with Improvement to Growth

This is very well organized and written paper. It was my pleasure to review this manuscript.

Critical review:

1.     Title of the manuscript should be more comprehensive.

2.     In context of global warming you haven't provided any information about reaction of the plants on biotic or abiotic stress. To tell the truth there is no information about volatile organic compounds. 

3.     The aim of the study must be defined precisely. What do you want to express by your manuscript?

4.     I don't understand Figure 1 included to the Results section.

5.     Figures 2 and 3 should be rather presented as supplementary material.

6.     I don’t see the conclusions. Personally I know the readers who pay attention only to the conclusions. Please do it.

Some other paper for consideration:

Sitophilus granarius responses to blends of five groups of cereal kernels and one group of plant volatiles

Journal of Stored Products Research 63: 63-66  (2015)

DOI: 10.1016/j.jspr.2015.05.007

Effect of phenolic acid content on acceptance of hazel cultivars by filbert aphid

Plant Protection Science 55(2): 116-122 (2019)

DOI: 10.17221/150/2017-PPS

Volatile organic compounds released by Rumex confertus following Hypera rumicis herbivory and weevil responses to volatiles

Journal of Applied Entomology 140(4): 308-316

DOI: 10.1111/jen.12238

Author Response

For clarity I included my responses after the reviewer's comments in boldfaced italics.

1. Title of the manuscript should be more comprehensive.  The title has been changed to be more comprehensive.

2. In context of global warming you haven't provided any information about reaction of the plants on biotic or abiotic stress. To tell the truth there is no information about volatile organic compounds. It was not the intent of this manuscript to address specific environmental stresses such as temperature extreme, drought, elevated carbon dioxide or exposure to other anthropogenic gases that relate to global warming. We merely mention the fact that wildfire events have increased in intensity which can be attributable to global warming trends. This paper specifically just targeted the effects of smoke and certain chemical constituents within smoke on the physiological responses of sunflower and its longer-term growth performance.

3. The aim of the study must be defined precisely. What do you want to express by your manuscript? The purpose of this work was clearly stated at the end of the Introduction (lines 97-104).

4. I don't understand Figure 1 included to the Results section. While we agree Figure 1 is more a Methods figure, highlighting the vascular connectivity of sunflower and the pathways 11C-photosynthates would follow as they transport to sink tissues, the authors feel it is necessary to post this figure early in the manuscript by placing it in the opening text of the Results. That way readers can understand and appreciate the nature of the radiographic images that follow in Figures 2 & 3. It’s unfortunate that the journal’s format places the Materials and Methods section later in the body of the manuscript.

5. Figures 2 and 3 should be rather presented as supplementary material. The authors disagree with this reviewer’s view. We feel these images provide reinforcing visualization of the changes in 11C-photosythate transport pathways and should remain in the main body of the manuscript. Also, we feel this heightens awareness of the radiotracer imaging technology.

6. I don’t see the conclusions. Personally, I know the readers who pay attention only to the conclusions. Please do it. A Conclusions section was added to the revised manuscript.

7. Some other papers for consideration:  We did not include the suggested papers numbered 1 & 3 as references in the manuscript because we did not see an appropriate connection with the purpose of the present work.  As stated earlier, the purpose of this work was to examine the influences of liquid smoke treatment and treatment with certain dihydroxybenzene constituents known to be present in smoke on the vascular connectivity in sunflower with studies examining the longer-term effects of chronic liquid smoke treatment on growth performance.

 Additional responses to suggested references below.

1. Sitophilus granarius responses to blends of five groups of cereal kernels and one group of plant volatiles Journal of Stored Products Research 63: 63-66 (2015) DOI: 10.1016/j.jspr.2015.05.007 How are the responses of granary weevils to blends of cereal VOCs relevant to the present work? We did not feel this reference was relevant to the present work.

2. Effect of phenolic acid content on acceptance of hazel cultivars by filbert aphid. Plant Protection Science 55(2): 116-122 (2019) DOI: 10.17221/150/2017-PPS This might be the only work in this group of references that has some relevance as it examined the allelopathic effects of foliar phenolics on aphid behavior. There could be some significance to sunflower elevation of foliar lignin content implying elevation of phenolic metabolism that would endow greater plant protection to herbivores though this would only be speculative.  Regardless, we included this reference to satisfy the reviewer.

3. Volatile organic compounds released by Rumex confertus following Hypera rumicis herbivory and weevil responses to volatiles. Journal of Applied Entomology 140(4): 308-316.  DOI: 10.1111/jen.12238 This reference is not relevant to the present work where it examined the effects of weevil herbivory on plant VOC release.